# Sleep Disturbances in Child and Adolescent Mental Health Disorders: A Review of the Variability of Objective Sleep Markers

**DOI:** 10.3390/medsci6020046

**Published:** 2018-06-04

**Authors:** Suman K. R. Baddam, Craig A. Canapari, Stefon J. R. van Noordt, Michael J. Crowley

**Affiliations:** 1Yale Child Study Center, Yale School of Medicine, New Haven, CT 06510, USA; stefonv0@gmail.com (S.J.R.v.N.); Michael.Crowley@yale.edu (M.J.C.); 2Division of Pulmonary Medicine, Department of Pediatrics, Yale School of Medicine, New Haven, CT 06510, USA; craig.canapari@yale.edu

**Keywords:** sleep, mental health, electroencephalography (EEG), children, adolescents, Attention Deficit Hyperactivity Disorder (ADHD), anxiety, autism, arousal

## Abstract

Sleep disturbances are often observed in child and adolescent mental health disorders. Although previous research has identified consistent subjective reports of sleep disturbances, specific objective sleep markers have not yet been identified. We evaluated the current research on subjective and objective sleep markers in relation to attention deficit hyperactivity disorders, autism spectrum disorders, anxiety and depressive disorders. Subjective sleep markers are more consistent than objective markers of actigraphy, polysomnography, and circadian measures. We discuss the causes of variability in objective sleep findings and suggest future directions for research.

## 1. Introduction

Significant neurobiological, physiological, and social changes occur during childhood and adolescence [1,2], including changes in circadian and sleep systems that regulate sleep duration and timing [3]. The duration of sleep decreases as development progresses, from about 14.5 h at 6 months of age to 8 h at age 16 years [4]. Other macrostructural changes, including changes in sleep stages, sleep architecture, and sleep efficiency, occur during childhood and adolescence [5]. Also, circadian changes, such as a shift toward later circadian chronotype and evening-type sleep patterns, emerge in adolescence [6]. These changes in sleep patterns across development, especially in the context of mental health problems, can be difficult for some youth. 

Along with changes in sleep patterns, many youth develop mental health disorders in childhood and adolescence [7]. Attention Deficit Hyperactivity Disorder (ADHD) (8.6%), mood disorders (3.7%) [8], and autism spectrum disorders (0.7%) (ASD) [9] are common mental health problems in children that show an onset in early childhood [10]. In adolescents, common mental health issues include anxiety disorders (31.9%), behavior disorders (19.1%), mood disorders (14.3%), and substance use disorders (11.4%) [11]. Further, child and adolescent mental health disorders are characterized by significant comorbidity with a wide range of severity [8,11,12].

Sleep difficulties in youth with mental health problems are common, including increased sleep latency, nocturnal awakenings, nightmares, snoring, restless sleep, excessive daytime sleepiness, bedtime struggles, and fear of dark [13]. Sleep problems are known to have complex bidirectional relationships with childhood psychiatric disorders [14]. Historically, the evidence linking subjective sleep reports with reliable objective sleep markers in youth with mental health disorders has been inconsistent. Although youth with anxiety, depression, ADHD, or ASD often report subjective sleep disturbances [15,16,17], there is a paucity of evidence on reliable objective markers of sleep in pediatric mental health disorders [17,18]. An earlier meta-analysis of children with ADHD identified sleep onset difficulties, bedtime resistance and difficulty with waking up in morning on subjective reports, high sleep onset latency and true sleep on actigraphy, and low sleep efficiency on polysomnography [19]. However, a recent meta-analysis only identified high percentage of stage 1 sleep as the significant finding in children with ADHD [20]. In major depressive disorder (MDD), a systematic analysis identified decreased sleep onset latency and rapid eye movement (REM) abnormalities as reliable sleep findings [21]. Objective sleep markers such as REM abnormalities, prolonged sleep onset latency, sleep fragmentation, and reduced sleep efficiency were identified in some studies of pediatric mental health disorders. However, several other studies have not shown any differences in objective sleep markers between children with mental health disturbances and healthy controls [22]. 

Herein, we review the literature on objective markers of sleep disturbances in the context of youth mental health disorders. We focused on current research studies and literature predominantly published during the last decade which includes objective measurements of sleep (polysomnography (PSG) or actigraphy) or circadian rhythms. Specifically, we examined studies in children and adolescents with anxiety disorders, depressive disorders, ADHD, and ASD diagnosed by diagnostic and statistical manual of mental disorders (DSM) criteria. We excluded studies of children with primary sleep disorders. From the selected studies, we identified objective sleep markers and their associations with subjective reports of sleep disturbances. 

## 2. Objective and Subjective Markers of Sleep Disturbances

Objective markers of sleep disturbances include measurements by PSG, actigraphy, and measures of circadian biology. Polysomnography, the most widely used and validated standard for the evaluation of sleep, is a multi-modal instrument which assesses a range of physiological changes in electroencephalography (EEG), respiration, and heart rate [23], with an established scoring method to identify macrostructural sleep characteristics [24,25]. Actigraphy, a wrist-based instrument that measures movement by an accelerometer, is another reliable tool for measuring sleep and wake rhythms [23,26]. Circadian rhythms are assessed using laboratory-based protocols [27], including assessment of melatonin and cortisol [28]. Arousals from sleep, defined as behavioral awakenings, an abrupt shift in the EEG frequency and desynchronization of EEG [29], have been examined in research studies as markers of sleep instability. Brief arousals from sleep, referred to as microarousals and cyclic alternating patterns, are used as markers of sleep disturbances [30,31] whereas subjective sleep patterns are assessed from several instruments including Children’s Sleep Habits Questionnaire [32], Adolescent Sleep-Wake Scale, Sleep Disturbances Scale for Children, Sleep Self-report, School Sleep Habits Survey [33], and sleep diaries.

### 2.1. Sleep in Anxiety and Depressive Disorders

Subjectively, parents of children with generalized anxiety disorder (GAD) report resistance to bedtime, delay in sleep onset, short sleep duration, high anxiety before sleep, and daytime sleepiness [34]. Similarly, adolescents with anxiety have high bedtime worries/fears, insomnia symptoms, daytime sleepiness on self-reports and high sleep onset latency, total sleep time and wake after sleep onset duration, and low sleep efficiency on sleep diaries [35]. In depression, children and adolescents typically report insomnia and, in some cases, hypersomnia, with severe depression associated with comorbid insomnia and hypersomnia [15]. In MDD, sleep diaries indicate subjective sleep complaints such as long sleep latency, a high number of awakenings, and high wake after sleep onset, although, these subjective reports are less common than in anxious children and adolescents [36]. Gender differences emerged in depression with females showing greater numbers of sleep complaints than males [15]. 

Recent actigraphy studies did not reveal any differences between children with GAD and healthy controls, despite reported subjective sleep disturbances [34]. However, a recent study in adolescents with GAD showed long sleep onset latencies, but unexpectedly greater sleep duration, compared to typically developing children [35]. Few studies have examined sleep patterns of children and adolescents with depression using actigraphy. Adolescent males with depression showed a shift toward later circadian phases from weekdays to weekends [37]. Short sleep duration and lower sleep efficiency were identified in adolescents with the seasonal affective disorder, a variant of depressive disorder [38].

Polysomnography in anxiety disorders showed reduced latency to REM [39], high sleep onset latency in children [36,39], low sleep efficiency and a high number of sleep arousals compared to controls. In multiple night PSG, youth with an anxiety disorder had greater sleep onset latency on the first night compared to the second night [36]. At-home PSG, in contrast to in-lab PSG, showed children with GAD had high sleep efficiency and fewer REM periods. The authors explained that children with anxiety had high bedtime resistance at home and went to bed later, which led to high sleep efficiency (from decreased sleep onset latency and decreased wake after sleep onset) [40]. In depression, low sleep efficiency, low proportion of slow wave sleep, and a high number of arousals have been observed [41] along with decreased total sleep time [42]. Interestingly, the dissipation of slow wave activity was slower and flatter [42], suggesting differences in the homeostatic dissipation of sleep pressure in children with depression. However, in the PSG study by Forbes et al., no differences were observed in depression when compared to healthy controls [36].

Lower nighttime cortisol, a marker of circadian and arousal differences, was observed in prepubertal children with anxiety [43], but no differences were observed in adolescents [44] or MDD [43]. However, adolescents with depression show higher peri sleep onset cortisol than healthy controls [44]. Additionally, arousal assessed in anxiety disorders showed pre-sleep arousal levels correlated with objective sleep findings—with pre-sleep somatic and cognitive arousal negatively correlated with REM sleep and total sleep time, respectively [40]. 

Eveningness chronotype assessed by chronotype questionnaires was consistently high in children with depression [45] and adolescent males [37] and was associated with the earlier development of depression symptoms [46] and circadian phase delay [47]. Later chronotype and social jetlag were observed in adolescent females with the seasonal affective disorder [38]. Circadian period, as measured by actigraphy, was found to be high in male and low in female children and adolescents with depression, in combination with low circadian amplitude [48]. 

### 2.2. Summary of Sleep in Anxiety and Depressive Disorders

In summary, children and adolescents with anxiety disorders show reliable subjective reports of greater sleep onset latency, bedtime fears, and greater wake after sleep onset duration [34,36]. Interestingly, findings on actigraphy do not show differences in sleep in children [34] but show high sleep onset latency and high sleep duration in adolescents with anxiety when compared to healthy controls [35]. PSG results corresponded with the findings of increased sleep onset latency, low sleep efficiency, and low slow wave sleep in anxiety and depression [36,41]. However, in a comparative study, sleep findings on PSG (sleep onset latency, sleep duration, sleep efficiency) were significantly greater in anxiety than depression and healthy controls, but similar in youth with depression and healthy controls [36]. Sleep efficiency was also inconsistent when assessed by PSG conducted at home versus the laboratory [34,40]. Cortisol level separated in some studies but not in others [43,44]. Chronotype differences, such as evening chronotype, phase delay, and social jet lag were consistently present in children and adolescents with depression [37,45,46]. The sleep findings of individual anxiety and depressive disorder studies are presented in Table A1 (Appendix A).

#### 2.2.1. Sleep in Attention Deficit Hyperactivity Disorder

In children with ADHD, subjective reports indicate a wide range of sleep problems, including difficulty with sleep onset, daytime sleepiness [49,50,51,52,53,54,55,56], anxiety prior to sleep [50,53,55,57], high severity of insomnia [50,53,55,56], high awakenings at night [50,51,53,56], difficulty waking up, less refreshing sleep [54], insufficient sleep [57], resistance to bedtime [51], and restless sleep [55]. However, a study by Mullin et al. identified no differences in subjective reports of sleep on sleep diary measures [58]. 

On actigraphy, children with ADHD have been identified to take longer to fall asleep [52,53,59], have lower sleep efficiency, lower sleep time [53,54], greater sleep fragmentation [59], and a high wake after sleep onset duration [59]. Greater day-to-day variability in sleep onset is present in youth with ADHD when compared to children with other psychiatric disorders and healthy controls [52]. However, sleep findings in actigraphy were not significantly different in other studies comparing children [60] and adolescents [58] with ADHD to healthy controls.

In-home PSG has shown that children with ADHD have short sleep duration, short REM sleep, a smaller percentage of REM sleep, and longer sleep onset latency [57]. However, in-lab studies of children with ADHD compared to healthy controls show greater REM duration [56,61], sleep period [61], and REM latency [49], as well as shorter total sleep time and less sleep in stage 1 and stage 3 [56]. These contrasting results are worth considering as no significant differences were present in other studies measuring overnight PSG [50,55,62] or Multiple Sleep Latency Test (MSLT—an objective assessment of sleepiness conducted with multiple naps during the day using EEG) [55]. High microarousals with increased motor activity during light and REM sleep was present in children with ADHD [63]. Cyclic alternating patterns, a marker of sleep instability and arousal, were found to be lower in children with ADHD, suggesting a state of hypoarousal in ADHD [49,64]. However, others have reported no differences in cyclic alternating patterns [62].

Cortisol levels vary across the day in children with ADHD, being lower in the morning and higher in the evening [65]. One study showed high urinary melatonin levels in children with ADHD [66], whereas another study did not show any differences in the melatonin levels in children with ADHD compared to typically developing children [67]. However, later melatonin onset at bedtime and earlier melatonin offset were present in children with ADHD [67]. Evening chronotype in children with ADHD was associated with resistance to bedtime [68] and delayed melatonin onset [69].

#### 2.2.2. Summary of Sleep in Attention Deficit Hyperactivity Disorder 

Overall, children with ADHD subjectively report sleep onset difficulties, daytime sleepiness, anxiety before sleep, and awakenings at night [50,53,55,57], corroborated on actigraphy by increased sleep onset latency, lower sleep time, and lower sleep efficiency [52,53,59]. However, at least one study did not find any differences between ADHD and typically developing children on actigraphy [58]. In PSG, longer sleep onset latency and shorter sleep duration were observed along with REM abnormalities in youth with ADHD [49,51,57]. In-lab vs. home PSG produced variable REM findings [57,61], whereas multiple studies did not identify objective sleep differences in PSG between youth with ADHD and healthy controls [50,55,62]. High melatonin and cortisol were identified [65,66], but the differences were not replicated in other studies [67]. Evening chronotype was common in ADHD and associated with resistance to bedtime [68,70]. The sleep findings of ADHD studies are presented in Table A2 (Appendix A).

### 2.3. Sleep in Autism Spectrum Disorders

Subjective reports of sleep in children with ASD include long mean sleep latency [71,72,73], short sleep duration, high nighttime awakenings, anxiety before sleep, and bedtime resistance [73]. Low functioning children with ASD are reported to have more severe sleep disturbances, including more frequent night awakenings, greater bedtime resistance, delay in sleep onset, later bedtimes and wake times, and less sleep than high functioning children with ASD [74]. However, a recent study did not find differences between typically developing children and children with ASD in subjective reports of sleep [72].

On actigraphy, preschool, school-aged children, and adolescents with ASD confirmed the subjective sleep findings, with long sleep latency, decreased sleep duration, and increased wake after sleep onset duration [73,75]. Other findings include low sleep efficiency in children [71,73,76] and adolescents [77] with ASD. Recent work also showed greater night-to-night variability in wake time, wake after sleep onset periods, and sleep efficiency in children with ASD [78], whereas greater variability in sleep onset latency has been reported in high functioning adolescents with ASD [76]. 

Polysomnography findings show short total sleep time, short REM latency [79], long sleep onset latency, less slow-wave sleep, low microarousals, and more sleep-wake transitions in children with ASD compared to typically developing children [72]. Low proportion of slow wave sleep and light sleep (high arousals) have been associated with high repetitive behaviors and poor social behaviors and intellectual measures [72]. Severe autism was linked to more pronounced sleep abnormalities with high sleep onset latency, high wake after sleep onset duration, low total sleep time, low slow wave sleep, and prolonged REM latency. Cyclic alternating patterns, the marker of arousal and sleep instability measured by visual inspection of sleep EEG, are greater among children with regressive autism compared to non-regressed children with autism and typically developing children [74]. Poor sleepers in autism have more affective problems, fewer social interactions, and longer sleep latency [80]. Moreover, youth with ASD also show low sleep efficiency, low sleep duration, and high variability in sleep latency from night 1 to 2 [80]. 

High salivary cortisol and overall blunted cortisol rhythms were identified in children with autism, with higher cortisol levels associated with severe symptoms of autism [81]. Melatonin secretion rate has been found to decrease in prepubertal children with autism [82], but not in adolescents [77]. Other studies did not identify differences in measures of cortisol or melatonin in children [83] or adolescents with autism [77].

### 2.4. Summary of Sleep in Autism Spectrum Disorders

Collectively, children and adolescents with autism commonly have reports of longer sleep latency, short sleep duration, and high resistance to bedtime on subjective measures [71,73]. Actigraphy validates the subjective findings of long sleep latency and short sleep time along with low sleep efficiency [71,73,77]. The PSG findings consisted of short total sleep time, short REM latency, low sleep efficiency, and low slow wave sleep [72,74,79], with severe autism associated with more severe sleep abnormalities as well as cyclic alternating patterns [72,74]. Melatonin and cortisol abnormalities were identified in children with autism but were not identified in adolescents [77,81,82]. The findings in actigraphy, PSG, and circadian measures were not consistently identified across studies. The sleep findings of autism spectrum disorder studies are presented in Table A3 (Appendix A).

## 3. Discussion

The goal of this review was to identify objective sleep characteristics associated with mental health disorders in children and adolescents. Collectively, the findings highlight that subjective reports of sleep problems are common in mental health disorders but do not necessarily coincide with reliable objective markers of sleep disturbances measured by actigraphy and PSG. High sleep onset latency, short sleep duration, and resistance to bedtime were commonly present in subjective measures of anxiety, ADHD, and ASD across diagnostic categories [34,35,49,57,72,73]. Although actigraphy showed high sleep onset latency [35,53,59], low sleep efficiency, and shorter sleep duration in ADHD and autism [71,73,75], these findings were not specific for a single disorder. Similarly, PSG showed long sleep onset latency, low sleep efficiency and low slow wave sleep as the common objective findings in autism [72,74,79], ADHD [56,57], anxiety, and depression [36,41], but were not specific for individual disorders and also varied across studies. However, severe autism was associated with more abnormal sleep findings [72,74,80]. The location of the study (at home vs. in-lab PSG) also produced variable findings in anxiety [34,40] and ADHD [57,61], and sleep findings also varied on multi-night PSG assessments [36,55]. Overall, specific macrostructural findings in mental health disorders were not identified. Nonspecificity of objective sleep findings in child and adolescent mental health disorders have been documented in the literature [13,14]. 

The overlapping objective sleep findings observed across child and adolescent mental health disorders suggest possible common sleep mechanisms. Child and adolescent mental health disorders often present with high rates of comorbidity [8,11]. When controlling for comorbid psychiatric symptoms, objective sleep markers show little differentiation in a study of sleep markers in ADHD [53]. In another study comparing diagnostic categories, child and adolescent psychiatric symptoms did not vary in sleep and cortisol measures [84], highlighting possible common pathways to objective sleep markers. However, severe mental health symptoms [72,74] and the presence of multiple comorbid mental health disorders have been associated with greater objective sleep disturbances [41]. Also, mental health symptoms, especially anxiety, present as temporary state and stable trait characteristics [85], both of which are known to affect sleep and cortisol measures [78,83] differentially. Delineating state and trait characteristics in mental health may be important for identifying sleep pathology. Considering the heterogeneity of state-trait characteristics and common sleep findings across diagnostic categories, identifying specific biobehavioral phenotypes based on biological phenotypes (such as attention, arousal, motivation) could assist with mapping reliable objective sleep markers across disorders and diagnostic categories [86,87]. 

Reliable macrostructural sleep findings have consistently eluded child and adolescent mental health disorders. Technical and methodological characteristics of actigraphy and PSG should be acknowledged when evaluating the findings of objective sleep markers. Actigraphy may overestimate sleep [88] during periods of inactivity. In actigraphy, greater night-to-night variability is found in ADHD [52,89] and autism [76,78], hence, an extended duration of measurement is needed to identify reliable results. Similarly, night-to-night variability is identified in PSG studies [90]. Therefore, single night PSG, commonly used in research studies [49,50,56,72,91], may be insufficient for identifying objective findings because of “first night effect”. Sleep findings often are different on the first night of sleep in the lab because of novel sleeping environment. “First night effect”, associated with high sleep latency, has been identified in children with ADHD [55,92] and autism [80] on multi-night studies. Given that sleep efficiency increases and wake after sleep onset decreases on the second night [93], multiple nights of PSG may be necessary to identify reliable objective differences in sleep [94]. The location of the sleep study (home vs. in-lab PSG) produces varying sleep findings in youth with anxiety [39,40] and ADHD [49,57], suggesting sleep environment moderates sleep phenotypes. The macrostructural characteristics of sleep identified with EEG have not revealed reliable differences in past studies of child and adolescent mental health disorders [13,14]. Macrostructural characteristics of sleep are assessed by visual scoring with rules that were originally established in 1968 [25]. Manual sleep scoring [24,25] can be problematic as it has low correspondence for delineating electrophysiological activity and does not take into account the temporal and spatial resolution or autonomic changes [95] during sleep. Examining the microarchitectural characteristics of sleep at greater temporal resolution using spectral methods [96] may help identify specific sleep findings in child and adolescent psychiatric disorders [97]. 

Arousals from sleep are an integral part of the sleep process [98] and are associated with measurable changes in the EEG [29]. Arousals associated with awakenings, microarousals (less than 3 s), and cyclic alternating patterns (CAP) that reflect sleep stage instability have been evaluated as objective markers of poor sleep [30,31]. High awakenings at night are common in anxiety [34,36], ADHD [50,55,57], and autism from subjective reports whereas arousals, microarousals and CAP are assessed by PSG and include EEG, behavioral, and autonomic components [30]. Frequent PSG arousals from sleep are identified in children with depression [41] and ADHD [63] and low microarousals in autism [72]. High cyclic alternating pattern, the marker of sleep instability and arousal [99], was reported in a regressive form of autism [74] with no differences in another study [79]. Low CAP in ADHD [49,64] is in line with evidence of a hypoarousal state in ADHD; however, other studies have failed to replicate this finding [62]. It is worth noting that microarousals and CAPs are manually scored, which likely contributes to their low reliability across studies. Computer-assisted techniques to identify EEG, respiratory and autonomic arousals during sleep in children and adolescents are likely to enhance the reliability of objective findings of sleep disruption [100].

Generalized CNS arousal, defined as optimal sensory, motor, and affective drive, is necessary for survival and interaction with the environment [101,102], and is, therefore, a critical biological process operating during sleep, wakefulness, and affective modulation. Hypothalamic-pituitary-adrenal (HPA) axis and the locus coeruleus-norepinephrine system are essential for the optimal functioning of the arousal system [103]. Dysregulation of arousal systems along with the HPA axis has been proposed in the pathophysiology of depression [104], autism [105] and ADHD [106]. Cortisol is an essential hormone of the HPA axis and was examined in several studies. High daytime cortisol has been reported in anxiety [43] and autism [81], and low daytime cortisol in ADHD [65], suggesting neuroendocrinal differences in sleep-wake systems. However, other studies have not corroborated these findings [43,44,77]. Methodological differences in measuring cortisol are present across studies which likely contributes to the inconsistent findings [107]. Heart-rate variability, a measure of autonomic arousal, has been observed to be higher in children with autism [91] and ADHD [108]. Research on arousal using reliable cortical and autonomic measures and methods in youth [109] are needed to identify more stable objective markers of sleep disturbances, particularly those that emphasize measurement reliability. 

## 4. Future Directions

We conducted this review to identify objective sleep markers associated with mental health disorders in children and adolescents. Several subjective sleep markers were found to be more consistent and common across child mental health disorders than objective sleep markers, which were not specific and varied across studies. A gap exists in identifying the pathophysiology of sleep disturbances and their links to mental health disorders in youth. Future research should focus on these links to clarify objective markers of sleep and their association with subjective reports and mental health symptoms. Also, mental health symptoms are heterogeneous and cross existing boundaries of syndromes, hence using biobehavioral phenotypes based on pathophysiology may help identify reliable objective markers of sleep [110]. It is essential to control for circadian, homeostatic, and environmental factors (light, physical activity) [111] that impact objective findings of sleep. Multimethod assessments using actigraphy, PSG, EEG, and subjective measures are needed [88] as single measurement methods identify specific markers unique to the method. Detailed analysis of EEG using spectral analysis may also reveal subtle findings not identified by manual scoring methods. Lastly, examining dysregulation and variability of arousal patterns within biobehavioral phenotypes across sleep and mental health may identify specific treatments and help advance the mission of precision medicine [86].

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
