# Peer review of "Sleep Disturbances in Child and Adolescent Mental Health Disorders: A Review of the Variability of Objective Sleep Markers"

_medsci, 2018, doi:10.3390/medsci6020046_

Round 1

Reviewer 1 Report

Authors have presented a detailed review of objective sleep markers in children with comorbid sleep and psychiatric disorders.

The detailed review and a very thoughtful discussion about the potential reasons for the variability in objective markers is admirable.

A summary table would enhance the readability and provide quick summary to the readers if authors would like to consider the addition.

Author Response

Authors have presented a detailed review of objective sleep markers in children with comorbid sleep and psychiatric disorders

The detailed review and a very thoughtful discussion about the potential reasons for the variability in objective markers is admirable

A summary table would enhance the readability and provide quick summary to the readers if authors would like to consider the addition

We thank the reviewer for the useful comments. We added the summary findings of individual studies in a tabular format in the supplementary materials. Table 1, Table 2 and Table 3 present the summary of individual studies in anxiety and depression, attention deficit hyperactivity disorder, and autism spectrum disorder and are attached after the references in the manuscript.

Reviewer 2 Report

Excellent review article and an important contribution to the field.

Author Response

Excellent review article and an important contribution to the field.

We would like the thank the reviewer for taking the time to review the article and the positive comments. 

Reviewer 3 Report

Sleep Disturbances in Child and Adolescent Mental Health Disorders: A Review of the Variability of  Objective Sleep Markers.

 Suman Baddam, Craig Canapari, Stefon van Noordt, and Michael J. Crowley

This is a very important topic for child psychiatrists that reviews our current state of knowledge about sleep characteristics in children and adolescents with metal disorders. The authors provided detailed critical analysis of published data on subjective and objective measures of sleep in pediatric patients with ADHD, mood and anxiety disorders and autism spectrum disorders, and propose future directives for research.   Article is very written and includes critical/main references. 

Comments:

Line 56.  “ Specifically, we examined studies in children and adolescents with anxiety disorders, depressive disorders, ADHD, and ASD [8] diagnosed by DSM” . Please clarify reference # 8 . I seems to unrelated to this sentence.  

Line 74. Since subjective sleep characteristics are being assessed using different instruments, I would recommended rewording the following sentence “whereas subjective  sleep patterns are assessed from Children’s Sleep Habits Questionnaire [29] and sleep diaries “  to include a broader definition of assessment tools.  

There are few important systemic reviews and Meta-Analysis studies that addressed subjective and objective characteristics of sleep in children with ADHD that are not mentioned in the manuscript. Please address this .

These are      the references : Díaz-Román A, Hita-Yáñez E,      Buela-Casal G. Sleep characteristics in children with attention deficit      hyperactivity disorder: systematic review and meta-analyses. J Clin Sleep      Med 2016;12(5):747–756.

Cortese      S, Faraone SV, Konofal E, Lecendreux M. Sleep in children with      attention-deficit/hyperactivity disorder: meta-analysis of subjective and      objective studies. J Am Acad Child Adolesc      Psychiatry. 2009;48:894–908. 

Author Response

We would like to thank the reviewer for taking the time to review the article and the useful comments to enhance the quality of the article.

Comment 1:

Line 56.  “ Specifically, we examined studies in children and adolescents with anxiety disorders, depressive disorders, ADHD, and ASD [8] diagnosed by DSM” . Please clarify reference # 8 . I seems to unrelated to this sentence.

We agree with the reviewer that the reference Merikangas et al does not add to the statement. We removed the reference. We changed the statement to “Specifically, we examined studies in children and adolescents with anxiety disorders, depressive disorders, ADHD, and ASD diagnosed by DSM”

Comment 2:

Line 74. Since subjective sleep characteristics are being assessed using different instruments, I would recommended rewording the following sentence “whereas subjective  sleep patterns are assessed from Children’s Sleep Habits Questionnaire [29] and sleep diaries “  to include a broader definition of assessment tools.

We agree with the reviewer about the availability of multiple self-assessment instruments. We modified the Line 74 to whereas subjective sleep patterns are assessed from several instruments including Children’s Sleep Habits Questionnaire [32], Adolescent Sleep-Wake Scale, Sleep Disturbances Scale for Children, Sleep Self-report, School Sleep Habits Survey [33], and sleep diaries”.

The following reference was added

1.

Erwin, A. M.; Bashore, L. Subjective Sleep Measures in Children: Self-Report. Front Pediatr 2017, 5, doi:10.3389/fped.2017.00022.

Comment 3:

There are few important systemic reviews and Meta-Analysis studies that addressed subjective and objective characteristics of sleep in children with ADHD that are not mentioned in the manuscript. Please address this

These are      the references : Díaz-Román A, Hita-Yáñez E,      Buela-Casal G. Sleep characteristics in children with attention deficit      hyperactivity disorder: systematic review and meta-analyses. J Clin Sleep      Med 2016;12(5):747–756

Cortese      S, Faraone SV, Konofal E, Lecendreux M. Sleep in children with      attention-deficit/hyperactivity disorder: meta-analysis of subjective and      objective studies. J Am Acad Child Adolesc      Psychiatry. 2009;48:894–908

Thank you for highlighting the important meta-analyses that we did not include in the introduction. We concur with the interviewer and include the systematic reviews in the introduction.

Line 49 – We add “An earlier meta-analysis of children with ADHD identified sleep onset difficulties, bedtime resistance and difficulty with waking up in morning on subjective reports, high sleep onset latency and true sleep on actigraphy, and low sleep efficiency on polysomnography [19]. However, a recent meta-analysis only identified high percentage of stage 1 sleep as the significant finding in children with ADHD [20]. In major depressive disorder (MDD), a systematic analysis identified decreased sleep onset latency and REM abnormalities as reliable sleep findings [21].”

The following references were added per the reviewer’s suggestion

Cortese, S.; Faraone, S. V.; Konofal, E.; Lecendreux, M. Sleep in children with attention-deficit/hyperactivity disorder: meta-analysis of subjective and objective studies. J Am Acad Child Adolesc Psychiatry 2009, 48, 894–908, doi:10.1097/CHI.0b013e3181ac09c9.

Díaz-Román, A.; Hita-Yáñez, E.; Buela-Casal, G. Sleep Characteristics in Children with Attention Deficit Hyperactivity Disorder: Systematic Review and Meta-Analyses. J Clin Sleep Med 2016, 12, 747–756, doi:10.5664/jcsm.5810.

Augustinavicius, J. L. S.; Zanjani, A.; Zakzanis, K. K.; Shapiro, C. M. Polysomnographic features of early-onset depression: A meta-analysis. Journal of Affective Disorders 2014, 158, 11–18, doi:10.1016/j.jad.2013.12.009.
